# Person-Centred Diabetes Care: Examining Patient Empowerment and Diabetes-Specific Quality of Life in Slovenian Adults with Type 2 Diabetes

**DOI:** 10.3390/healthcare12090899

**Published:** 2024-04-26

**Authors:** Tina Virtič Potočnik, Nina Ružić Gorenjec, Matic Mihevc, Črt Zavrnik, Majda Mori Lukančič, Antonija Poplas Susič, Zalika Klemenc-Ketiš

**Affiliations:** 1Primary Healthcare Research and Development Institute, Community Health Centre Ljubljana, Metelkova ulica 9, SI-1000 Ljubljana, Slovenia; nina.ruzic.gorenjec@mf.uni-lj.si (N.R.G.); matic.mihevc@gmail.com (M.M.); crt.zavrnik@zd-lj.si (Č.Z.); majda.mori.lukancic@gmail.com (M.M.L.); antonija.poplas-susic@zd-lj.si (A.P.S.); zalika.klemenc-ketis@zd-lj.si (Z.K.-K.); 2Department of Family Medicine, Faculty of Medicine, University of Maribor, Taborska ulica 8, SI-2000 Maribor, Slovenia; 3Community Health Centre Slovenj Gradec, Partizanska 16, SI-2380 Slovenj Gradec, Slovenia; 4Institute for Biostatistics and Medical Informatics, Faculty of Medicine, University of Ljubljana, Vrazov trg 2, SI-1000 Ljubljana, Slovenia; 5Department of Family Medicine, Faculty of Medicine, University of Ljubljana, Poljanski nasip 58, SI-1000 Ljubljana, Slovenia

**Keywords:** quality of care, patient empowerment, health-related quality of life, diabetes mellitus type 2, integrated care, primary care

## Abstract

Patient empowerment is crucial for promoting and strengthening health. We aimed to assess patient empowerment and diabetes-specific health-related quality of life (HRQoL) in adults with type 2 diabetes (T2D). A multi-centre, cross-sectional survey was conducted among adults with T2D in urban and rural primary care settings in Slovenia between April and September 2023. The survey utilised convenience sampling and included sociodemographic and clinical data, the Diabetes Empowerment Scale (DES), and the Audit of Diabetes-Dependent QoL (ADDQoL). The study included 289 people with T2D and a mean age of 67.2 years (SD 9.2). The mean overall DES score was 3.9/5 (SD 0.4). In a multivariable linear regression model, higher empowerment was significantly associated with residing in a rural region (*p* = 0.034), higher education (*p* = 0.028), and a lack of comorbid AH (*p* = 0.016). The median overall ADDQoL score was −1.2 (IQR [−2.5, −0.6]). The greatest negative influence of diabetes on HRQoL was observed in the domain ‘Freedom to eat’, followed by ‘Freedom to drink’, ‘Leisure activities’, and ‘Holidays’. Despite high empowerment among adults with T2D, the condition still imposes a personal burden. Integrated primary care models should prioritise the importance of implementing targeted interventions to enhance diabetes empowerment, address comorbidities, and improve specific aspects of QoL among individuals with T2D.

## 1. Introduction

Type 2 diabetes (T2D) remains one of the most common diseases in the modern world, with its prevalence steadily increasing in recent years [1]. It represents a major challenge for future health policy, in terms of both its socioeconomic impact and health-related quality of life (HRQoL) [2].

The intractable and chronic nature of the disease, the complexity of treatment and the multitude of daily decisions that people with T2D make for themselves (including dietary choices, physical activity, and the regular monitoring of blood pressure and glucose levels) together contribute to the difficulty of managing T2D [3,4]. The Chronic Care Model, a highly recognised and recommended model for improving the quality of diabetes care [5], strongly emphasises person-centredness and self-management support. Power sharing between patients and healthcare providers is vital for high-quality care with an informed, empowered patient and a prepared, proactive, multidisciplinary healthcare team [6,7]. Such integrated care leads to positive clinical, educational, psychosocial, and behavioural outcomes where the empowered patient becomes an equal member of a multidisciplinary team [3,4,7].

Patient empowerment is one of the fundamental principles of promoting and strengthening health [7]. It is defined as a process whereby patients have the knowledge, skills, attitudes, and self-awareness necessary to influence their own behaviour and that of others to improve their quality of life [8]. Patient empowerment not only refers to those psychological aspects related to motivation, behavioural change, and adherence to self-care strategies but also involves a process of psychological transformation and acceptance in which the reality of illness becomes part of one’s identity [9].

The positive impact of patient empowerment on clinical and psychosocial outcomes is well established [10,11,12,13,14]. Past research indicates that factors such as higher education [15,16,17], diabetes-specific knowledge [15,18,19], improved glycaemic control [4,16,20,21,22], and community support [4] are linked to higher levels of empowerment in individuals with T2D. Conversely, anxiety [23,24], depression [25], and older age [15,17,19,21,26] show negative associations with self-efficacy and patient empowerment. 

Slovenia, a high-income central European country, reports an estimated 9.3% prevalence rate with 145,000 cases of adult-onset diabetes [27]. While integrated care has been successfully implemented at the Slovenian primary healthcare level, numerous challenges persist in achieving structured collaboration and providing self-management support to empower patients [28,29,30]. Furthermore, patients have emphasised the difficulties associated with the intricate task of accepting and managing their illnesses, along with taking responsibility for their health. Addressing workforce shortages and burnout, along with improving accessibility for vulnerable populations, remain ongoing hurdles in scaling up integrated primary care [30]. The only study conducted in 2012 to assess diabetes-specific HRQoL in elderly Slovenian individuals with T2D revealed that this chronic illness imposes a substantial personal burden, leading to noticeable impacts on dietary choices, dependence on others, and family life [31].

Understanding an individual’s empowerment in managing diabetes can assist in driving suitable recommendations for personalised interventions tailored to their needs [20]. The aim of the present study was to evaluate patient empowerment and explore its relationship with various sociodemographic and health factors while also assessing diabetes-specific HRQoL in Slovenian adults with T2D receiving integrated care.

## 2. Materials and Methods

### 2.1. Study Design

We conducted a multi-centre, cross-sectional study among the Slovenian adult population with T2D. This cross-sectional study was designed and conducted in accordance with the Strengthening the Reporting of Observational Studies in Epidemiology (STROBE) guidelines. We followed the STROBE checklist to ensure comprehensive reporting and enhance the transparency and reproducibility of the study [32].

### 2.2. Study Setting

The study was conducted across multiple primary healthcare centres (PHCs) in Slovenia. In selecting the PHCs for inclusion in the study, we established baseline criteria to ensure the representation of both urban and rural settings in Slovenia. The criteria included considerations such as the population size, geographic location, and economic contribution to the national gross domestic product (GDP). PHC Ljubljana is situated in the capital city, catering to a substantial population of approximately 300,000 residents. It distinctly embodies the characteristics of an urban setting, contributing 38.4% of the total GDP of Slovenia in 2022. Conversely, PHC Slovenj Gradec, PHC Trebnje, and PHC Zreče serve an estimated population of 50,000 residents in Eastern Slovenia, representing rural regions. Their respective contributions to the national GDP in 2022 ranged from 2.6% to 6.5% [33].

### 2.3. Healthcare Context

The Slovenian health system operates under the combination of the Beveridge and Bismarck models, ensuring equitable healthcare access [34]. At the primary healthcare level, a capitation system is established with family physicians acting as gatekeepers. They serve as central coordinators of care, collaborating horizontally with various healthcare professionals and vertically connecting with specialists at secondary and tertiary levels, as well as the community [35].

### 2.4. Integrated Primary Care Model

This research was conducted within an integrated primary care model in Slovenia. In Slovenian family medicine practices, a team consisting of a family physician, a practice nurse, and a registered nurse offers integrated care with a standardised approach for patients with chronic diseases like T2D. This approach covers screening, treatment, education, and quality control. During regular check-ups, the physician assesses and adjusts the treatment plan, while the registered nurse screens for complications, evaluates the patient’s psychosocial status, and provides education on non-pharmacological measures [34]. Family medicine practices regularly collaborate with teams at health education and promotion centres staffed by registered nurses, physiotherapists, dietitians, and psychologists [35], as well as community nurses, who serve as a bridge between physicians and patients [29].

#### 2.4.1. Study Population and Patient Recruitment

This research, undertaken between April and September 2023, employed a convenience sampling methodology. Patients were conveniently recruited from family medicine practices by their family physicians during regular appointments at PHCs. Recruitment continued until the predetermined sample size required to validate the Slovenian version of the Diabetes Empowerment Scale (DES), which was the objective of separate research, was attained. In the present study, our primary objective was to assess patient empowerment with the additional aim of evaluating diabetes-specific HRQoL. Consequently, the participants were asked to also complete the Audit of Diabetes-Dependent QoL (ADDQoL) questionnaire.

#### 2.4.2. Inclusion and Exclusion Criteria

The inclusion criterion was a confirmed diagnosis of T2D according to the guidelines [7] with a minimum duration of 1 year. 

The exclusion criteria included the following: type 1 diabetes or gestational diabetes, <18 years of age, and a documented diagnosis of cognitive decline obtained from the participant’s medical records. This diagnosis was based on comprehensive assessments of the individual’s clinical presentation, medical history, and relevant test results conducted by family physicians and other healthcare professionals. Cognitive decline presents a significant comorbid condition that can substantially impact patient empowerment, disease management, and outcomes [36]. Considering these complexities, we excluded patients with documented cognitive decline from the study to ensure the homogeneity of the study population and minimise confounding factors. By focusing on individuals with confirmed diagnoses of T2D without cognitive impairment, we aimed to obtain a clearer understanding of the specific factors influencing diabetes management and outcomes in this population.

### 2.5. Data Collection

The participants provided all data through a structured, self-administered questionnaire.

#### 2.5.1. Sociodemographic and Clinical Characteristics

The questionnaire included information on the participants’ age, gender, zip code of permanent residence, education, employment status, marital status, years since diagnosis of T2D, treatment method for T2D, presence of arterial hypertension (AH) as a comorbidity, anthropometric measurements (height and weight), and glycated haemoglobin (HbA1c, also self-reported) information.

#### 2.5.2. Diabetes Empowerment Scale (DES)

The DES, developed by Anderson et al., is a 28-item, self-reported instrument measuring the psychosocial self-efficacy of individuals living with diabetes, exhibiting a Cronbach’s alpha coefficient of 0.96 [37]. 

As the original DES is in English, we undertook translation, adaptation, and validation procedures to create the Slovenian version of the DES (SL-DES). The translation and adaptation of the DES into Slovenian followed the World Health Organization and the International Society for Pharmacoeconomics and Outcomes Research (ISPOR) guidelines [38]. The steps included the following: (a) obtaining permission from the original author of the scale to translate the DES [37] into Slovenian, (b) forward translation of the scale into Slovenian by two independent experts, (c) the reconciliation of the forward translations into a single forward translation, (d) the back translation of the scales into English by two bilingual experts, (e) the harmonisation of the new translations with each other and with the initial version, (f) the proofreading and finalisation of the SL-DES.

SL-DES demonstrated excellent internal consistency, evidenced by a Cronbach’s alpha coefficient of 0.90. The SL-DES encompasses three distinct subscales: ‘Managing Psychosocial Aspects of Diabetes’ (9 items, α = 0.84), ‘Dissatisfaction and Readiness to change’ (9 items, α = 0.64), and ‘Setting and Achieving Diabetic Goals’ (10 items, α = 0.87). Responses on the scale range from 1 to 5, with higher scores indicating greater empowerment.

#### 2.5.3. Audit of Diabetes-Dependent Quality of Life (ADDQoL)

To assess diabetes-specific HRQoL among Slovenian adults with T2D, we employed the already-validated Slovenian version of the ADDQoL instrument with Cronbach’s alpha of 0.93. The questionnaire includes two overview items to offer single-item indicators of HRQoL. Generic Quality of Life (GQoL) gauges respondents’ feelings about their present QoL, with scores ranging from +3 (excellent) to −3 (extremely bad). Diabetes-Dependent Quality of Life (DDQoL) prompts patients to evaluate what their HRQoL would be if they did not have diabetes, with scores ranging from −3 (very much better) to 1 (worse). The impact of diabetes is then measured in 19 life domains. Participants rate each domain on two scales: ‘Impact Ratings’, ranging from −3 to +1, and ‘Importance Ratings’ from 0 to +3. The domain’s score is a product of these two scales (i.e., each domain is evaluated based on two questions). An average weighted impact score (AWI) is then calculated, providing an overall assessment of the impact of diabetes on HRQoL with a range from −9 to +3. Lower AWI scores indicate poorer HRQoL [39].

### 2.6. Sample Size Calculation

There is no universally accepted standard for determining an appropriate sample size for validation studies, with the literature suggesting a patient-to-item ratio between 5 and 20 [40]. In our research, we followed a widely accepted guideline for sample size calculation, recommending a minimum of 10 participants for each item on the scale for a validation study, resulting in an ideal of respondent-to-item ratio of 10:1 [41]. Since the Diabetes Empowerment Scale (DES) consists of 28 items, our target was to recruit at least 280 participants for the study.

### 2.7. Statistical Analysis

Numerical variables were summarised using means and standard deviations (SDs) or medians and interquartile ranges (IQRs) in the case of asymmetrical distribution. Categorical variables were presented with absolute and relative frequencies. The association of DES with other variables was explored using a multivariable linear model in which all measured variables were included with some categorical variables dichotomised and BMI categorised. Specifically, the included covariates were gender, age (years), region of residence (rural/urban), education (dichotomised into higher vs. primary school or secondary/vocational school), marital status, employment status, time since diagnosis of T2D (years), treatment with antihyperglycemics, treatment with insulin, comorbidity of T2D and AH, HbA1c (%), and BMI (categorised into underweight (BMI < 18.5), normal weight (18.5 < BMI < 24.9), overweight (25 < BMI < 29.9), and obese (BMI > 30), and there were no underweight participants in our sample). The association of DES and ADDQoL was assessed using Pearson’s correlation coefficient. A *p*-value of <0.05 was considered statistically significant. 

In calculating the overall DES with its subscales and ADDQoL, we addressed missing values by considering three versions. We calculated the score of a participant only if (i) all questions were answered, (ii) at least one question was answered, and (iii) at least 50% of corresponding questions were answered. The results were similar across all three versions. To include more participants and consider only the more reliable respondents at the same time, we opted for version (iii), for which the results are reported. Additionally, we also report the number of respondents with complete answers. Furthermore, upon careful examination, we found that there was no discernible pattern in the missing data. 

All analyses were carried out with the R programming language, version 4.3.2 [42].

## 3. Results

### 3.1. Sociodemographic and Clinical Characteristics

Table 1 presents sociodemographic characteristics of the 289 participants with a mean age of 67.2 years (SD 9.2), of whom 157 (54.5%) were women. Most participants (72.1%) were residing in a rural setting, 72 participants (25.7%) had an education level higher than secondary school, and 147 (51.2%) had arterial hypertension as a comorbidity. Furthermore, 231 (82.0%) were classified as overweight or obese, with a BMI exceeding 25 kg/m^2^.

### 3.2. Assessment of Patient Empowerment 

Measurements of patient empowerment are presented in Table 2. The mean overall DES score was 3.9 (SD 0.4). The means for the three DES subscales were as follows: 3.9 (SD 0.5) for subscale I, ‘Managing the Psychosocial Aspect of Diabetes’, 3.8 (SD 0.5) for subscale II, ‘Assessing Dissatisfaction and Readiness to Change’, and 4.0 (SD 0.5) for subscale III, ‘Setting and Achieving Diabetes Goals’ (see also Figure 1).

The multivariable analysis of variables associated with DES revealed that empowerment is significantly higher for individuals with T2D from rural regions (*p* = 0.034), those with higher education (*p* = 0.028), and those without comorbid AH (*p* = 0.016) (Table 3).

Additionally, a comparison of the DES and ADDQoL questionnaires revealed the absence of a correlation between DES and ADDQoL scores (r = −0.1, *p* = 0.078), indicating that these two scales measure different aspects related to diabetes (Figure 2).

### 3.3. Assessment of Diabetes-Specific Health-Related Quality of Life

Measurements of diabetes-specific HRQoL are presented in Table 2. The overall ADDQoL score, calculated for 286 participants, ranged from −8.8 to 0.9 within a predefined range of −9 to +3. The median for the overall ADDQoL score was −1.2 (IQR [−2.5, −0.6]). Additionally, the first two overview items were included in the analysis. The median for the GQoL item was 1 (i.e., ‘quite good’) with an IQR of [1, 2]. For the DDQoL item, the median was −1 (IQR [−2, −1]), suggesting that participants believed their QoL would be ‘a little better’ if they did not have T2D (Table 2).

For all 19 life domains of the ADDQoL questionnaire, at least 75% of participants achieved scores of at most zero (Figure 3). The lowest negative scores were observed for the domain ‘Freedom to eat’, with a median score of −2 (IQR [−4, −1]), which was followed by ‘Freedom to drink’, ‘Leisure activities’, and ‘Holidays’, all with a median score of −2 and an IQR of [−4, 0]. The least negative influence of T2D on HRQoL was observed for the domain ‘People’s reaction’, for which 82% of participants had a score of 0. This was followed by the domains ‘Physical appearance’, ‘Dependence on others’, ‘Financial situation’, ‘Self-confidence’, ‘Sex life’, and ‘Friendship and social life’, all with a median score of 0 and an IQR of [−2, 0].

## 4. Discussion

This study provides comprehensive insights into the prevailing status of patient empowerment and diabetes-specific HRQoL among Slovenian adults with T2D. The results indicated a high level of participants’ empowerment, which was significantly associated with higher education, residing in rural regions, and the absence of AH as a comorbidity. Additionally, the study underscores the negative influence of T2D on the HRQoL, highlighting the burden and barriers individuals face due to this chronic disease. Specifically, the participants reported that T2D had the most negative influence on the freedom to eat and drink as desired and on leisure activities and holidays.

The participants in our study demonstrated high overall DES scores, consistent with prior research [16,20], including the original DES validation study [37], and surpassing many other foreign studies [15,17,43,44,45]. The recent World Health Organization report highlights Slovenia’s well-functioning integrated, person-centred primary health care [46]. This structure and interprofessional collaboration [34,35] encourage the active involvement of individuals with T2D in decision making about their health, further enhancing and promoting patient empowerment.

The findings indicated that the DES subscale ‘Setting and Achieving Diabetes Goals’ exhibited a slightly higher mean score than the other two subscales, aligning with the results from previous studies [16,17,43]. This particular subscale evaluates patients’ perceived self-efficacy in identifying relevant and achievable diabetes goals, as well as overcoming the barriers to their achievement [37]. Structured goal setting is the best way to aid individuals with T2D in setting behaviour goals to practise healthy lifestyles and improve glycaemic control [16]. Since the early 2000s, health education/promotion centres have been implemented at all PHCs across Slovenia, ensuring uniform and accessible health promotion and comprehensive and structured health education while aiming to identify and mitigate regional disparities in care [35].

Our research demonstrated a significant association between higher DES scores and residing in rural settings. This may be attributed to close-knit communities, which could contribute to a heightened sense of social support and a higher level of empowerment [4,12]. However, other studies have shown that primary care is evenly spread across socioeconomic classes in Slovenia [35], the lack of an association between diabetes-specific HRQoL and one’s place of living [31], and similar levels of diabetes knowledge among the elderly in urban and rural areas, implying that the accessibility of healthcare services and diabetes education programmes does not differ across Slovenia [47]. 

Establishing strong community and social support networks for persons with T2D is crucial [4]. To enhance patient empowerment levels and improve their HRQoL, it is essential to shift and down-step care from healthcare professionals to patients and informal caregivers. Power sharing between patients and healthcare providers is vital for high-quality care with an informed, empowered patient and a prepared, proactive, multidisciplinary healthcare team [6]. This approach has been implemented through telemedicine [48] and peer support [49] pilot studies in Slovenian primary care.

This study has shown that an education level higher than secondary school is significantly associated with empowerment in people with T2D, aligning with previous research [4,15,16,17,43]. Higher levels of education are often associated with increased health literacy. This connection could contribute to a better understanding of diabetes-related processes and an increased awareness of the importance of self-care in disease management to prevent or delay severe complications [19].

In our study, there was no significant correlation between diabetes empowerment and the age of participants. Previous studies have shown an inverse association between age and empowerment [15,17,19,21,26], which could be explained by lower levels of health literacy observed in older persons with T2D. On the contrary, research by Simonsen et al. showed that adults with T2D, aged 27–57 years, had a lower level of empowerment, lower emotional well-being, more psychosocial distress, and less diabetes-related social support than older participants [4]. 

Our findings indicate no significant correlation between patient empowerment and the duration of T2D, as well as between patient empowerment and glycaemic control. Previous studies have shown that individuals diagnosed with T2D for a longer duration demonstrated higher levels of empowerment [17,43], reflecting the prolonged learning and acquisition of skills and knowledge through experience and exposure to diabetes education. This increased empowerment is associated with the ability to make better decisions for self-care, set targets, and achieve goals [16]. Previous studies consistently show that individuals perceiving higher levels of empowerment tend to exhibit greater success in self-care, embracing a healthy lifestyle that contributes to improved glycaemic outcomes [4,16,20,21,22].

In terms of diabetes-specific HRQoL, our study’s overall ADDQoL score ranked among the least negative in the published literature [50,51,52]. The comparison of the generic items, GQoL and DDQoL, revealed alignment with an Australian study [50] in which participants similarly rated their present HRQoL on average as ‘quite good’ and believed it would be, on average, ‘a little better’ if they did not have T2D. Due to unavailable calculations in other studies [31,51,52], a comparison was not feasible. 

We found the greatest negative influence of diabetes on HRQoL observed in the life domain ‘Freedom to eat’, which aligns with prior research findings [31,51,52]. Diabetes necessitates dietary restrictions and the constant monitoring of food intake, exercise, glucose, and blood pressure levels to decrease the risk of complications and improve health and QoL [51]. Furthermore, the most substantial decrease in HRQoL was experienced among obese Slovenian elderly individuals with T2D [53,54]. The life domain least affected by diabetes was ‘People’s reaction’, which also aligns with findings from other studies [31,51,52]. The reasons behind this observation may be multifaceted and influenced by a combination of individual, social, and cultural factors.

Our findings suggest that the present diabetes-specific HRQoL among Slovenian adults surpasses that in the solely published Slovenian research by Turk et al. over a decade ago [31], as the overall median ADDQoL score in our study was higher (−1.2 compared to −1.6). Turk et al. classified the overall ADDQoL score into two groups based on the first quartile (i.e., −3), resulting in 25.3% of participants in the lower QoL group. In our study, the same cut-off value yielded only 17.1% of participants in the lower QoL group. This implies an improvement in the HRQoL of people with T2D over the last decade, possibly influenced by the implementation of a person-centred, integrated primary care model of family medicine practices in 2011 [34,35]. This comparison requires careful interpretation due to differences in study populations, as the study by Turk et al. focused exclusively on an elderly population with an age range of 65 to 84 years, while our study included adults with a mean age of 67.2 years (SD 9.2), ranging from 30 to 91 years. However, despite these age variations, many other studies did not find associations between age and diabetes-specific HRQoL [31,50,51]. 

Surprisingly, our study did not reveal any correlation between DES and ADDQoL scores, indicating that these two questionnaires likely capture different concepts related to T2D, which aligns with findings from a study in Spain [19]. Changes in how empowered a person feels about managing their diabetes might not correlate directly with changes in their reported diabetes-specific HRQoL. While DES evaluates specific aspects of patient empowerment, guiding patients effectively, it does not assess empowerment comprehensively, covering aspects such as knowledge, skills, critical thinking, autonomy, abilities, values, and attitudes [55]. Moreover, concerns are raised about the correct interpretation and difficulties in comparing the results of the frequently and internationally known ADDQOL questionnaire [39].

### 4.1. Implication for Practice

The findings of this study underscore the significance of implementing targeted interventions to enhance patient empowerment, address comorbidities, and improve specific aspects of HRQoL among persons with T2D. The continuous promotion and implementation of an integrated primary care model would facilitate a person-centred approach that considers individual needs and challenges, enabling healthcare professionals to effectively support people with diabetes in achieving optimal health outcomes and overall well-being.

It is imperative for all healthcare providers to incorporate assessments of self-rated health, including measures of patient empowerment and HRQoL, into daily routine care. As DES and ADDQoL likely capture different aspects related to T2D, employing a complementary assessment for a comprehensive understanding of persons’ experiences and needs in diabetes management is necessary. 

Efforts should be directed towards enhancing empowerment among individuals with T2D, tailoring education and support according to their educational backgrounds and geographical settings. Additionally, proactive steps should be taken to address prevalent comorbidities such as AH, which can impede empowerment initiatives. 

Health professionals should acknowledge that age and the duration of T2D may not directly correlate with diabetes empowerment levels, and they should focus on providing ongoing, personalised education and support. They should recognise personal burdens and focus on continuous improvement in diabetes-specific HRQoL, as evidenced by the improvement over the last decade in Slovenia, possibly influenced by Slovenia’s well-functioning integrated, person-centred primary health care [34,35,46].

### 4.2. Strengths and Limitations

To our knowledge, this study is the first to assess patient empowerment among adults with T2D in Slovenia. It provides a thorough investigation of patient empowerment and diabetes-specific HRQoL among adults with T2D within a primary healthcare context. This study’s strength is grounded in its comprehensive, multi-centre methodology, which includes a comparison between urban and rural contexts to capture diverse healthcare dynamics and challenges facing residents in different regions of Slovenia. Furthermore, our research expands knowledge by demonstrating the nuanced relationship between empowerment and diabetes-specific HRQoL in individuals managing T2D, shedding light on areas requiring targeted intervention.

However, our study faced several limitations. Firstly, the cross-sectional nature of the study design and dependence on convenience sampling may have introduced a selection bias. Additionally, the cross-sectional design precluded the establishment of a cause-and-effect relationship between patient empowerment and other factors, allowing only the identification of association. Secondly, there might be a potential response bias in reporting empowerment and HRQoL. Since person-reported outcome measures are subjectively assessed through participants’ self-reports, their responses might be influenced by social desirability or other factors that could influence the overall DES and ADDQoL scores. Thirdly, our multivariable model accounted for only a small proportion of the total variation in the outcome, underscoring the necessity for additional variables to capture the complexity of patient empowerment more effectively. The existing literature indicates that patient empowerment is strongly influenced by individual behaviours, beliefs, attitudes, and values [8,9]. For a more comprehensive understanding, it is crucial to consider the impact of mental health, unmet needs, and the identification of individuals’ diabetes-related emotional distress and to assess the psychological adjustment to diabetes when measuring empowerment [55]. Lastly, there were some missing values in the DES and ADDQoL questionnaires, but upon careful examination, we found that there was no discernible pattern in the missing data.

## 5. Conclusions

Our findings highlight a high level of patient empowerment among Slovenian adults with T2D, which was notably associated with higher education, residing in rural areas, and the absence of AH as a comorbidity. The study also underscores the negative impact of T2D on their HRQoL, particularly in domains related to dietary freedom and leisure activities, emphasising the personal burden faced by individuals with this chronic illness. Understanding an individual’s empowerment level in disease management is crucial for tailoring person-centred care. Further longitudinal studies are essential for a deeper understanding of the behavioural, biological, and psychosocial factors related to patient empowerment. This knowledge can assist clinicians and policymakers in identifying high-risk groups, allocating resources effectively, and targeting evidence-based interventions to enhance self-management, improve patient empowerment, and address HRQoL challenges among individuals with T2D. 

## Figures and Tables

**Figure 1 healthcare-12-00899-f001:**
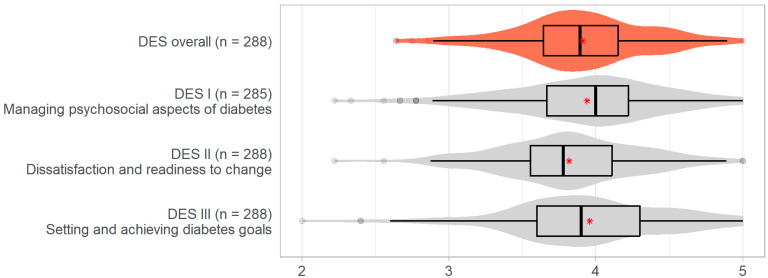
Distributions of overall DES and its subscales: boxplots in black, violin plots in red or grey, and means are marked with a red asterisk. Abbreviations: DES, Diabetes Empowerment Scale; DES I, subscale for Managing Psychosocial Aspects of Diabetes; DES II, subscale for Dissatisfaction and Readiness to Change; DES III, subscale for Setting and Achieving Diabetes Goals.

**Figure 2 healthcare-12-00899-f002:**
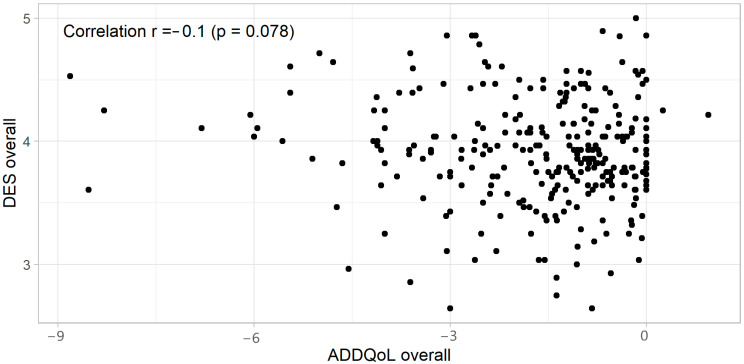
A scatter plot presenting the values of the overall DES and overall ADDQoL. Abbreviations: DES, Diabetes Empowerment Scale; ADDQoL, Audit of Diabetes-Dependent Quality of Life.

**Figure 3 healthcare-12-00899-f003:**
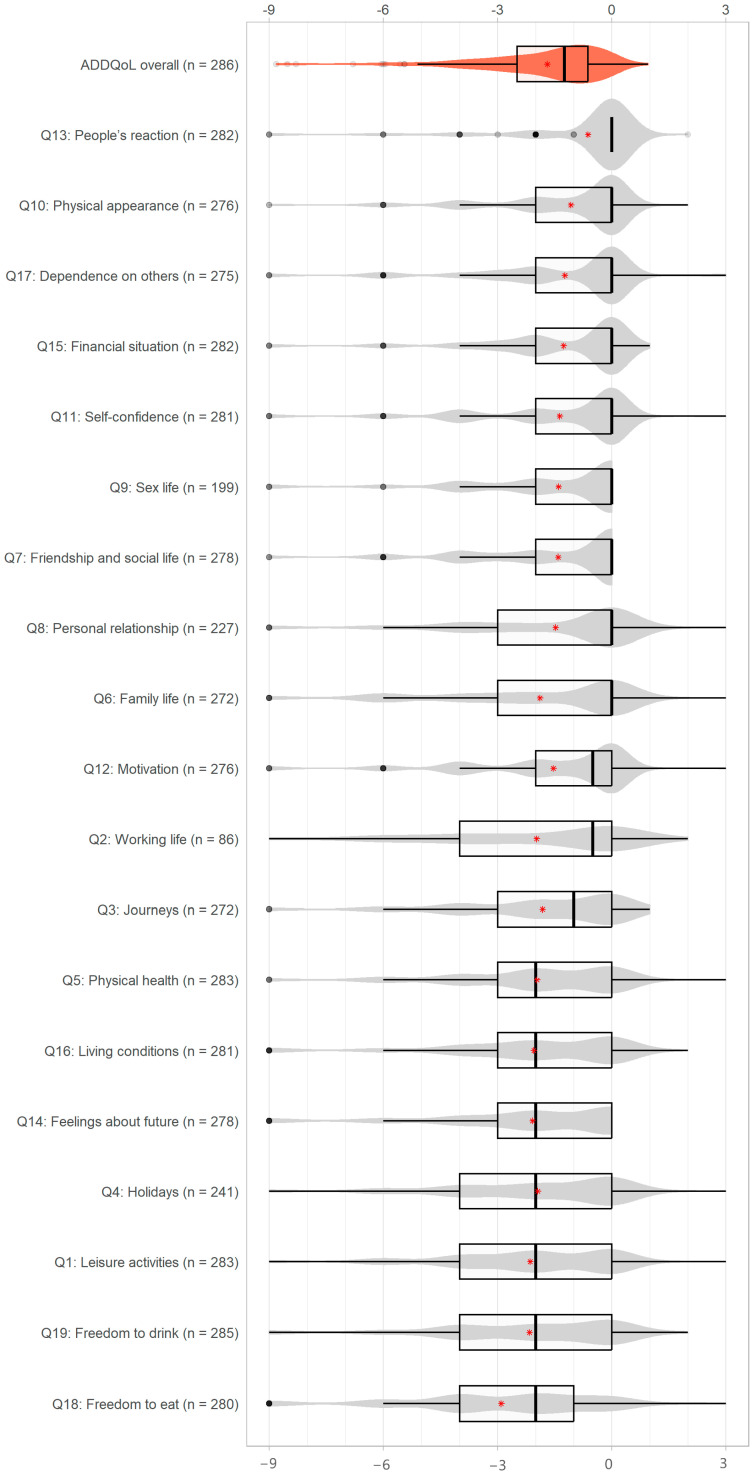
Distributions of the overall ADDQoL and all its 19 domains: boxplots in black, violin plots in red or grey, and means are marked with a red asterisk (where domains are ordered first by median, then by the first quartile, and lastly by their mean). Abbreviations: ADDQoL, Audit of Diabetes-Dependent Quality of Life.

**Table 1 healthcare-12-00899-t001:** Sociodemographic and clinical characteristics of 289 participants.

Characteristic	*n* (%)/Mean (SD)	[Range]
Age (years), *n* = 287	67.2 (9.2)	[30, 91]
Gender, *n* = 288		
Male	131 (45.5)	
Female	157 (54.5)	
Region of residence, *n* = 287		
Urban setting	80 (27.9)	
Rural setting	207 (72.1)	
Education, *n* = 280		
Primary school	40 (14.3)	
Secondary/vocational school	168 (60.0)	
Higher vocational college	55 (19.6)	
University education	12 (4.3)	
Master’s/doctoral degree	5 (1.8)	
Marital status, *n* = 284		
Married	193 (68.0)	
Single	18 (6.3)	
Divorced	15 (5.3)	
Widowed	58 (20.4)	
Employment status, *n* = 288		
Employed	52 (18.1)	
Unemployed	8 (2.8)	
Retired	228 (79.2)	
Comorbidity of T2D and AH, *n* = 287	147 (51.2)	
Time since diagnosis of T2D (years), *n* = 278, median [IQR]	8.0 [4.0–15.0]	[1, 34]
Treatment method for T2D		
Antihyperglycemics, *n* = 288	190 (66.0)	
Insulin treatment, *n* = 286	72 (25.2)	
Height (cm), *n* = 282	169.7 (9.4)	[141, 193]
Weight (kg), *n* = 282	83.4 (15.5)	[52, 135]
BMI (kg/m^2^), *n* = 282	28.9 (4.5)	[20.8, 45.1]
Underweight (BMI < 18.5)	0 (0)	
Normal weight (18.5 < BMI < 24.9)	51 (18.1)	
Overweight (25 < BMI < 29.9)	142 (50.4)	
Obese (BMI > 30)	89 (31.6)	
HbA1c (%), *n* = 224	7.3 (1.1)	[5.4, 13]

Abbreviations: *n*, number; SD, standard deviation; IQR, interquartile range; T2D, type 2 diabetes; AH, arterial hypertension; BMI, body mass index; HbA1c, glycated haemoglobin.

**Table 2 healthcare-12-00899-t002:** Diabetes Empowerment Scale with subscales and Audit of Diabetes-Dependent Quality of Life: descriptive statistics.

Variable	*n* Complete	*n* Considered	Mean (SD)	Median [IQR]	Range
DES	268	288	3.9 (0.4)	3.9 [3.6, 4.2]	[2.6, 5]
DES I	280	285	3.9 (0.5)	4.0 [3.7, 4.2]	[2.2, 5]
DES II	275	288	3.8 (0.5)	3.8 [3.6, 4.1]	[2.2, 5]
DES III	282	288	4.0 (0.5)	3.9 [3.6, 4.3]	[2, 5]
ADDQoL	241	286	−1.7 (1.6)	−1.2 [−2.5, −0.6]	[−8.8, 0.9]
GQoL	286	286	1.2 (0.9)	1 [1, 2]	[−3, 3]
DDQoL	286	286	−1.2 (0.9)	−1 [−2, −1]	[−3, 1]

Abbreviations: *n* complete, the number of respondents with all corresponding answers; *n* considered, the number of respondents considered in the calculation of a score (i.e., with at least 50% of corresponding questions answered); SD, standard deviation; IQR, interquartile range; DES, Diabetes Empowerment Scale; DES I, subscale for Managing Psychosocial Aspects of Diabetes; DES II, subscale for Dissatisfaction and Readiness to Change; DES III, subscale for Setting and Achieving Diabetes Goals; ADDQoL, Audit of Diabetes-Dependent Quality of Life; GQoL, General Quality of Life (feelings about one’s current QoL); DDQoL, Diabetes-Dependent Quality of Life (what HRQoL would be like if participants did not have diabetes).

**Table 3 healthcare-12-00899-t003:** Multivariable linear model for association of Diabetes Empowerment Scale with sociodemographic and clinical characteristics: adjusted R2 = 0.06, model *p* = 0.028.

Variable	Regression Coefficient (95% CI)	*p*
Female gender	0.06 [−0.06, 0.19]	0.321
Age (years)	0.00 [−0.01, 0.01]	0.986
Rural setting	0.15 [0.01, 0.29]	**0.034**
Higher education	0.16 [0.02, 0.30]	**0.028**
Marital status		0.182
Single vs. married	0.03 [−0.22, 0.27]	0.825
Divorced vs. married	−0.32 [−0.60, −0.03]	0.031
Widowed vs. married	−0.04 [−0.21, 0.13]	0.622
Employment status		0.432
Unemployed vs. employed	−0.27 [−0.68, 0.14]	0.198
Retired vs. employed	−0.05 [−0.26, 0.16]	0.621
Time since diagnosis of T2D (years)	0.01 [−0.00, 0.02]	0.099
Treatment with antihyperglycemics	−0.10 [−0.23, 0.03]	0.140
Treatment with insulin	−0.05 [−0.22, 0.12]	0.535
Comorbidity of T2D and AH	−0.16 [−0.29, −0.03]	**0.016**
HbA1c (%)	0.00 [−0.06, 0.06]	0.992
BMI		0.607
Overweight vs. normal	−0.09 [−0.27, 0.09]	0.319
Obese vs. normal	−0.06 [−0.25, 0.12]	0.502

Abbreviations: T2D, type 2 diabetes; BMI, body mass index; HbA1c, glycated haemoglobin; CI, confidence interval. The bold font was used in cases where the *p*-value was <0.05, indicating a statistically significant result

## Data Availability

The original data presented in the study are openly available via Harvard Dataverse at https://doi.org/10.7910/DVN/ZNQFGN (dataset published online on 25 March 2024).

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
