# Peer review of "Person-Centred Diabetes Care: Examining Patient Empowerment and Diabetes-Specific Quality of Life in Slovenian Adults with Type 2 Diabetes"

_healthcare, 2024, doi:10.3390/healthcare12090899_

Round 1
Reviewer 1 Report (New Reviewer)
Comments and Suggestions for Authors
The authors present a comprehensive analysis of patient empowerment and quality of life (QoL) concerning Type 2 Diabetes (T2D) among adults in Slovenia.
Their special interest was the factors predicting a high level of Empowerment. Furthermore, the study highlights the negative influence of T2D on HRQoL, emphasizing the challenges and barriers individuals face due to this chronic disease.
My observations and reservations regarding this report are as follows:
The study underscores the importance of patient empowerment in managing T2D and improving QoL. It also highlights the need for a supportive community and social networks, as well as the role of education in enhancing patient empowerment. The findings can inform strategies to improve diabetes care and patient outcomes in Slovenia and other similar settings.
- Introduction.
- Please mind that throughout the manuscript the use of the term HRQOL is inconsistent; I would suggest that the authors use the acronym HRQOL, when/if this was measured as opposed to general QOL.
- Similarly, in my opinion, the use of Empowerment should be preceded by PATIENT – especially in the abstract.
- In diabetology specifically, the term patient should be replaced by a person with Diabetes (PwD)- (Dickinson et al., 2017)
- Methods:
- Please define whether HbA1c was self-reported by the patients or reported precisely by the investigating personnel.
- Results:
- Table legends should be understandable by themselves without looking at the text body. Consider adding a short description of what each score indicates.
- Discussion
I could argue that the findings about health care accessibility in Slovenia – since according to NIJZ (Nation Institute of Public Health) data it does differ, however, the Authors argued well according to the cited literature.
Comments on the Quality of English Language
Author Response
"Please see the attachment."

Reviewer 2 Report (New Reviewer)
Comments and Suggestions for Authors
Person-Centred Diabetes Care: Examining Empowerment and 2 Diabetes-Specific Quality of Life in Slovenian Adults with 3 Type 2 Diabetes
The objective of the study was to evaluate quality of life in adults diagnosed with type 2 diabetes. The study included 289 adults diagnosed with type 2 diabetes, with an average age of 67.2 years. The study suggests that living with the condition still imposes a personal burden. Integrated primary care models are recommended to prioritize targeted interventions aimed at enhancing diabetes empowerment, addressing comorbidities, and improving specific aspects among individuals with T2D.
1. Authors should properly disclose the STROBE guidelines utilized in the cross-sectional study to avoid any methodological error.
2. Baseline criteria for the selection of the primary healthcare centers (PHCs) included in the study should be included to prove the rigor and reproducibility of the data acquired.
3. Proper methodology section for the adopted integrated care model for patient recruitment, patient demographics, data collection, and analysis procedures should be included.
4. For the present study, patients with cognitive decline is completely excluded. Inclusion and exclusion section will be properly included and discussed by explaining how each co-morbid condition aggravate the existing pathology.
5. How the questionnaire is conducted? The process of translation, adaptation, and validation of the Slovenian versions, should be properly disclosed in the methodology.
6. Sample size per group is limited to a minimum ratio of 10 participants per item. Authors should indicate the statistical variability in response patterns for each results with respect to the number of subjects.
8. Additional details on potential confounders or covariates adjusted for in the multivariable linear model should be disclosed in methodology.
9.Potential conflicts or limitations of the study by considering the missing patterns in the study subjects, would give additional rigor for the methodology.
10. Bioinformatics or software tools implemented for the study will be properly discussed in the methodology.
Author Response
Please see the attachment.

This manuscript is a resubmission of an earlier submission. The following is a list of the peer review reports and author responses from that submission.
Round 1
Reviewer 1 Report
Comments and Suggestions for Authors
The study was conducted soundly probably and article write good.
